# Biochemical, Structural Analysis, and Docking Studies of Spiropyrazoline Derivatives

**DOI:** 10.3390/ijms23116061

**Published:** 2022-05-27

**Authors:** Angelika A. Adamus-Grabicka, Mateusz Daśko, Pawel Hikisz, Joachim Kusz, Magdalena Malecka, Elzbieta Budzisz

**Affiliations:** 1Department of Bioinorganic Chemistry, Faculty of Pharmacy, Medical University of Lodz, Muszynskiego 1, 90-151 Lodz, Poland; 2Department of Inorganic Chemistry, Faculty of Chemistry, Gdansk University of Technology, Narutowicza 11/12, 80-233 Gdansk, Poland; mateusz.dasko@pg.edu.pl; 3Department of Molecular Biophysics, Faculty of Biology and Environmental Protection, University of Lodz, Pomorska 141/143, 90-236 Lodz, Poland; pawel.hikisz@biol.uni.lodz.pl; 4Institute of Physics, University of Silesia, 75 Pulku Piechoty 1, 41-500 Chorzow, Poland; joachim.kusz@us.edu.pl; 5Department of Physical Chemistry, Faculty of Chemistry, University of Lodz, Pomorska 163/165, 90-236 Lodz, Poland; magdalena.malecka@chemia.uni.lodz.pl; 6Department of the Chemistry of Cosmetic Raw Materials, Faculty of Pharmacy, Medical University of Lodz, Muszynskiego 1, 90-151 Lodz, Poland

**Keywords:** flavonoid derivatives, anticancer properties, spiropyrazoline, PARP1, DNA interaction

## Abstract

In this study, we evaluated the antiproliferative potential, DNA damage, crystal structures, and docking calculation of two spiropyrazoline derivatives. The main focus of the research was to evaluate the antiproliferative potential of synthesized compounds towards eight cancer cell lines. Compound **I** demonstrated promising antiproliferative properties, especially toward the HL60 cell line, for which IC_50_ was equal to 9.4 µM/L. The analysis of DNA damage by the comet assay showed that compound **II** caused DNA damage to tumor lineage cells to a greater extent than compound **I**. The level of damage to tumor cells of the HEC-1-A lineage was 23%. The determination of apoptotic and necrotic cell fractions by fluorescence microscopy indicated that cells treated with spiropyrazoline-based analogues were entering the early phase of programmed cell death. Compounds **I** and **II** depolarized the mitochondrial membranes of cancer cells. Furthermore, we performed simple docking calculations, which indicated that the obtained compounds are able to bind to the PARP1 active site, at least theoretically (the free energy of binding values for compound **I** and **II** were −9.7 and 8.7 kcal mol^−1^, respectively). In silico studies of the influence of the studied compounds on PARP1 were confirmed in vitro with the use of eight cancer cell lines. The degradation of the PARP1 enzyme was observed, with compound **I** characterized by a higher protein degradation activity.

## 1. Introduction

Despite huge advances in medicine, neoplastic diseases are a common cause of death in the human population. In the last decade, great attention has been focused on PARP1 inhibitors. Poly-ADP-ribose polymerases (PARPs) belong to enzymes that participate in several processes that are a vital for every living cell [1]. The PARP enzyme is activated in the event of DNA damage, the appearance of chronic pathological processes leading to metabolic disorders, cardiovascular diseases, or neurodegenerative changes [2]. The greatest attention was devoted to research on the influence of PARP on the treatment of neoplastic diseases. The aim of the first clinical trials of PARP inhibitor monotherapy was to evaluate the effectiveness of treatment in patients with breast and ovarian cancer. After the use of the PARP inhibitor, olaparib was evaluated positively, and studies were conducted on the efficacy of treatment of sporadic neoplasms (sporadic ovarian cancer and triple-negative breast cancer) [3,4]. The activation of PARP enzymes by the presence of DNA damage is known to trigger poly-ADP-ribosylation in various proteins that are crucial for DNA repair, presentation of genome integrity, regulation of transcription, proliferation, and apoptosis. The PARP polymerase enzyme plays a number of important roles in cells [5]. The PARP protein modulates the structure of chromatin, affects the metabolism, stability, and repair of DNA, and regulates gene transcription. It is involved in the detection of both single- and double-stranded DNA breaks. Current cancer treatments are largely based on the exposure of cancer cells to compounds that cause DNA damage. The limiting factor in effective chemotherapy is the high efficiency of the mechanisms of DNA damage repair of neoplastic cells. Cutting-edge research focuses on the search for compounds that are active against proteins involved in all DNA repair systems. High hopes for anticancer therapies are associated with PARP inhibitors. New methods are sought in order to improve the effectiveness of cancer treatment. One of these is therapy with PARP enzyme inhibitors. The synergism between PARP inhibitors and genotoxic chemotherapy or radiation therapy determined another direction of research on the application of these drugs [6,7]. In recent years, research has proven that PARP inhibitors are effective in breast and ovarian cancers [8]. PARP inhibitors prevent cancer cell multiplication by detecting defects in DNA structure and repairing the damage. The anticancer activity of PARP inhibitors can be enhanced by synergistic combination with other compounds with antitumor activity [9]. In the treatment of breast cancer, a PARP inhibitor was used in combination with an initial kinase inhibitor; in the treatment of ovarian cancer, doxorubicin was used to enhance the therapeutic effect of the PARP inhibitor; and in the treatment of pancreatic cancer, the effect of cisplatin with a PARP inhibitor was investigated [10]. Currently, three PARP inhibitors are used to treat recurrent ovarian cancer: olaparib [11], rucaparib [12], and niraparib [13]. These PARP inhibitors are used as anticancer drugs in the maintenance treatment of low-grade recurrent serous ovarian cancer and in the treatment of breast cancer in patients with BRCA1 or BRCA2 mutations. The most commonly used is olaparib, which blocks the PARP catalytic center and stabilizes PARP–DNA complexes [14]. As a result, it contributes to an increase in cytotoxicity, leading to the death of cancer cells. In clinical trials, the pairing of PARP inhibitors with different cytostatic drugs is expected for the treatment of cancer that is recurrent and resistant to previous treatments [15,16].

We have been conducting research on the biological properties of pyrazolines condensed with a chromone ring substituted with various functional groups for a long time [17,18]. It has been confirmed that flavone derivatives exhibit a number of biological properties [19,20] due to the presence of heteroatoms in their structure; however, our team pays greatest attention to anticancer properties. On account of the combination of the pyrazoline core containing nitrogen atoms with flavonoid derivatives, the biological potential of the compounds is greater [21]. Over several decades, many scientific papers have been created based on the research carried out on the properties of flavonoid compounds [20,22,23]. Scientists have paid much attention to anticancer and antioxidant activities [24]. Despite many documented scientific reports, further research is being conducted on the biological properties of these compounds. In our previous papers, we reported on the synthesis and biological evaluation of derivatives of 3-benzylidenechromanones and their spiropyrazoline analogues [17,25]. The aim of the research previously conducted was the synthesis of 3-benzylideneflavanones and 3-benzylidenechromanones with various substituents attached in the ortho, meta, or para positions in a condensation reaction between 2-phenylchroman-4-one or chroman-4-one, and the appropriate aryl aldehyde, in piperidine medium. The next step included the preparation of spiro-1-pyrazolines via the reaction of 3-benzylidene analogues with an ethereal solution of diazomethane in anhydrous acetone. The obtained compounds were examined for biological activity in four cancer cell lines: HL60 (human leukemia cell line), NALM-6 (human peripheral blood leukemia cell line), WM115 (melanoma cell line), and COLO205 (human colon adenocarcinoma cells). Furthermore, we tested their compatibility with blood cells, the effect on the integrity of the erythrocyte membrane, and the lipophilicity parameter, using the RP-TLC method. All synthesized compounds exhibited promising antiproliferative properties, but the best IC_50_ values were demonstrated by 3-arylidenechromanone derivatives with the diethylamine group. Among the spiro-1-pyrazoline analogues, the highest anticancer activity, especially toward leukemia cell lines, was presented by the spiroflavanones with a methoxy group substituted at the *meta* and *para* positions [17]. These compounds also induced mitosis block in G2/M. These results confirm that the tested compounds presented a strong influence on the cell cycle and arrest the HL60 cells in the G2/M phase. Encouraged by the results of previous studies, we decided to expand our research on the biological activity of compounds with a methyl group at the *meta* and *para* positions of the phenyl ring attached to the pyrazoline heterocyclic ring of spiropyrazoline.

In this article, we report the biochemical studies of 5′-(3-methylphenyl)-2-phenyl-4′,5′-dihydro-4*H*-spiro[chromano-3,3′-pirazol]-4-one (compound **I**, Figure 1) and 5′-(4-methylphenyl)-2-phenyl-4′,5′-dihydro-4*H*-spiro[chromano-3,3′-pirazol]-4-one (compound **II**, Figure 1). The synthesized compounds were subjected to biological tests that evaluated cytotoxic activity using the MTT method in eight cancer cell lines: HEC-1-A, Ishikawa, MCF7, HCC38, HL60, WM115, NALM-6, and COLO205. The results of cytotoxic activity in the last four cancer cell lines were published in a previous article [17]. Investigation into mitochondrial potential was performed using a JC-1 fluorescent probe, DNA damage was determined by a comet assay, PARP-1 degradation was assessed with a commercial kit, and necrotic cells were determined with fluorescent dyes. Subsequent studies on the crystal structure of spiropyrazoline derivatives and simple docking calculations indicate that the obtained compounds are able to bind to the PARP1 active site, at least theoretically.

## 2. Results and Discussion

### 2.1. Molecular and Crystal Structures

The molecular crystal structure of compound **I** and **II** is shown in Figure 2. Both compounds crystalize in a monoclinic system in the space group P2_1_/c P2_1_/n. The main skeleton of the examined structures consists of a chroman-4-one moiety (a benzene ring fused with a tetrahydropyran-4-one ring). At the C2 atom, the benzene ring is substituted. Furthermore, *p*,*o*-methylophenyl-4,5-dihydro-3*H*-pyrazole is attached at position C3. The pyran rings adopt a distorted envelope conformation with puckering ring parameters of QT = 0.416(3) Å, θ = 129.2(4)°, and φ2 = 252.4(5)° for compound **I**, and a conformation between an envelope and a half-chair with puckering ring parameters of QT = 0.4288(12) Å, θ = 128.13(17)°, and φ2 = 259.3(2)° for compound **II** [26]. The five-membered (pyrazole) rings adopt an envelope conformation in both cases, with puckering ring parameters of QT = 0.252(3) Å and φ2 = 292.0(7)° for compound **I**, and QT = 0.1973(13) Å and φ2 = 288.1(3)° for compound **II**. These conformations were confirmed by the following asymmetry parameters: ΔCs = 5.5(3)°, ΔCs = 2.2(3)° (compound **I**; 6-membered, 5-membered ring), and ΔCs = 10.46(12)°, ΔCs = 0.55(2)° (compound **II**; 6-membered, 5-membered ring) [27]. This agrees with our previously reported structure for similar compounds [17,25]. The benzene ring at the C2 atom is nearly perpendicular to the chroman moiety; the dihedral angles between the best planes through the benzene and chroman moiety rings are 82.36(11)° and 85.58(5)°. As shown in Figure 3, the overall molecular structures do not differ significantly.

#### Structure of Compound **I** and **II**

The solid-state structures of both compounds include a combination of C–H…O, C–H…N, and C–H…π interactions. As a matter of interest, we studied the type of substructure generated by each type of hydrogen bond that builds the three-dimensional framework. The geometry parameters for the hydrogen bonds are given in Table 1. In the crystal structure of compound **I** atom C5 acts as donor in C5–H5…O4^i^ hydrogen bonds (symmetry code (i): −x, −y, −z), forming the R_2_^2^(10) graph-set motif [28] (Figure 4a).

The second substructure is chain C(7) along the [010] direction formed by a C25–H25…O1^ii^ hydrogen bond (symmetry code (ii): −x, y − 1/2, −z + ½) (Figure 4b). The N2 atom is an acceptor for C6–H6…N2^iii^ (symmetry code (iii): x − 1, y, z) and C27–H27A…N2^iv^ (symmetry code (iv): x, −y + 1/2, z + ½) hydrogen bonds, which are produced by both C(8) chains along the [100] and [001] direction, respectively (Figure 4c,d). In addition, the crystal lattice is stabilized by C22–H22…Cg(1), related by the symmetry 1 − x, y + 1/2, 1/2 − z interaction with the geometrical parameters H22...Cg(1) 2.61 Å, C22…Cg(1) 3.553 Å, and <C22-H22…Cg1 172° (Figure 4e), producing a chain along the [010] direction.

In the case of the crystal structure of compound **II**, molecules create the chains C(6), C(8), and C(4), propagating along the [010], [100], and [010] directions, with the respective hydrogen bonds: C12–H12B…O1^v^, C24–H24…O4^vi^, and C11–H11…N1^v^ (symmetry codes (v): 1/2 − x, −1/2 + y, 1/2 − z; (vi): 1 + x, y, z ) (Figure 5a,b). Two hydrogen bonds, C12-H12B…O1^v^ and C11-H11…N1^v^, form the ring R_2_^2^(8) (Figure 5a). The C–H…π interaction is also observed in the crystal lattice, and C7 acts as a donor in the C7–H7…Cg(2) interaction connecting two molecules related by the inversion center (Figure 5c).

### 2.2. Molecular Docking

To verify that compounds **I** and **II** are able to bind to the PARP1 active site effectively, molecular docking studies were performed. The X-ray structure of PARP1 was retrieved from the Protein Data Bank (Protein Data Bank accession code 4HHY) and properly prepared for docking calculations. The docking procedure of the optimized ligands was performed using AutoDock Vina 1.1.2 software (Molecular Graphics Laboratory, The Scripps Research Institute, La Jolla, CA, USA). The calculated results for the proposed structures of compound **I** and **II** indicated that they could, at least theoretically, efficiently bind to the active site of the PARP1 enzyme. The free binding energies of both compounds were at the satisfactory level (Table 2); however, the free energy of binding calculated for compound **I** (−9.7 kcal mol^−1^) was lower than the free energy of binding calculated for compound **II** (−8.7 kcal mol^−1^). The free energy of binding values for compounds **I** and **II** were comparable to the free energy of binding calculated for quercetin (−8.9 kcal mol^−1^), which was used as a reference. The slightly lower free energy of binding of compound **I** indicates that the presence of the methyl group at the *meta* position of the terminal aromatic ring connected to the pyrazolidine unit may be more favorable in binding than its presence at the *para* position. Therefore, potentially higher PARP1 inhibitory activity of compound **I** than of compound **II** may be expected; however, biological evaluation should be performed in order to confirm this hypothesis.

The visualization of the putative binding modes for compounds **I** and **II**, and quercetin, using VMD 1.9 (University of Illinois at Urbana-Champaign, Urbana, IL, USA) is shown in Figure 6. It was noticed that all three compounds occupy the same region of the enzyme’s binding pocket. In the case of compound **I** (Figure 7), it can be seen that the tetracyclic core is well accommodated in the active PARP1 site, and is surrounded by some hydrophobic amino acid residues. The methyl group of compound **I** is in close proximity to the Ile211 and Leu216 residues (3.64 and 3.90 Å, respectively), suggesting the presence of hydrophobic interactions. In the case of compound **II**, such hydrophobic interactions between the *para*-substituted methyl group and hydrophobic amino acid residues were not detected. Furthermore, it can be observed that the aromatic units of compound **I** may create π–π interactions with the residues Tyr235 (3.42) and Tyr246 (3.54 and 3.64 Å), which can additionally stabilize the ligand–enzyme complex. All of the detected interactions may provide strong binding and promising inhibitory activity to compound **I**.

### 2.3. Biological Assay

#### 2.3.1. Cytotoxic Activity

In the first stage of the cytotoxicity of the study, the compounds were tested for cytotoxicity against four cancer cell lines: estrogen-responsive breast cancer cells MCF7, estrogen-unresponsive breast cancer cells HCC38, Ishikawa, and endometrium adenocarcinoma HEC-1-A. In addition, 4-chromanone and cisplatin were included in the tests as reference compounds. Table 3 lists the IC_50_ concentrations obtained for the entire series of compounds tested. The cytotoxicity results of the compounds analyzed for the COLO205, HL60, NALM-6, and WM115 cell lines were published in our earlier article [17].

The obtained results indicate that the analyzed compounds are cytotoxically active against all four cancer cell lines in the concentration range 10–120 µM. However, it is worth emphasizing that the antitumor activity of the compounds against the MCF7, HCC38, Ishikawa, and HEC-1-A lines is noticeably weaker than for the cancer lines HL60, NALM-6, WM115, and COLO205. The calculated IC_50_ values were in the range of ~27–55 µM. The HEC-1-A and MCF7 lines remained the most sensitive to the cytotoxic effects of the analyzed compounds.

The evaluation of the cytotoxicity of compounds against eight cancer cell lines served as an initial selection of compounds for further studies on the molecular mechanisms of their antitumor activity. On the basis of the calculated IC_50_ concentrations, the compounds with the highest activity against a particular cancer cell line were selected. We used 50 µM as the cutoff concentration. Based on the IC_50_ cytotoxicity results for the MCF7, HEC-1-A, HL60, and NALM-6 lines, both compounds **I** and **II** were tested. For the Ishikawa, WM115, and COLO205 lines, only the biological activity in compound **I** was analyzed, while for HCC38, only activity in compound **II** was analyzed.

#### 2.3.2. Analysis of DNA Damage Using the Alkaline Version of the Comet Assay (Single Cell Electrophoresis)—DNA Comet Assay

To assess DNA damage of cancer cells exposed to the compounds analyzed, the analysis of alkaline version of the comet assay was performed (pH > 13). The method allows one to specify both single- and double-strand DNA breaks and damage to DNA nucleotides caused by the negative effects of various factors. The results obtained from single cell electrophoresis show that the compounds damaged the DNA of all cancer cells used in the experiment (Figure 8). This DNA-damaging activity contributed to the cytotoxic effect of the investigated compounds. DNA damage of cancer cells exposed to spiropyrazoline derivatives ranged from ~10–25% depending on the cell line used. Interestingly, NALM-6 cells of acute lymphoblastic leukemia were clearly the most resistant to the genotoxic effects of the analogues based on the spiropyrazolines analyzed. DNA damage to NALM-6 cells after treatment with compound **I** and **II** was 12% and 9%, respectively. On the other hand, a high level of DNA damage was observed after treatment of MCF7, HCC38, and especially HEC-1-A cells with compound **II**. The level of DNA damage was ~20–23%.

Exemplary images of cells from the analysis of DNA damage by the alkaline comet assay (single cell electrophoresis) are shown in Figure 9. Supercoiled loops of DNA linked to the nuclear matrix of untreated control cells are seen to retain a compact structure resembling a comet head. Staining with DAPI fluorescent dye (4′,6-diamidin-2-phenylindole) emits very bright and intense fluorescence. In the case of cells exposed to genotoxic compounds, where DNA damage is observed, DNA supercoils are more relaxed, and thus migrate behind the head, forming a comet tail with weaker fluorescence.

#### 2.3.3. Determination of Apoptotic and Necrotic Cell Fractions by Fluorescence Microscopy (Double Staining of Cells with Fluorescent Dyes Hoechst 33258 and Propidium Iodide)

Referring to studies of the cytotoxic activity of the analyzed compounds and the determined IC_50_ values (µM), a microscopic analysis of the physiological state (viability) of cancer cells treated with the test compounds was performed. To analyze viability, cells were stained with Hoechst 33258 and propidium iodide fluorescent dyes after 24 h incubation with compounds at the calculated IC_50_ concentration. The obtained microscopic images confirm the cytotoxic and proapoptotic activity of spiropyrazoline-based analogues. The treated cancer cells were microscopically evaluated and representative areas were photographed at 150× magnification.

After incubation of cells with the compounds to be analyzed, mostly cells with intense blue fluorescence were observed in the microscopic field of view after staining with fluorescent dye. Microscopic observations indicated that cells treated with spiropyrazoline-based analogues were entering the early phase of programmed cell death. In addition, a small percentage of cells were observed in the late phase of apoptosis (purple fluorescence). It is worth noting that after exposure of the cells to the tested compounds, only a small percentage of necrotic cells (intense red fluorescence) was observed. Exemplary images of the determination of the apoptotic and necrotic cell fractions by fluorescence microscopy are shown in Figure 10.

#### 2.3.4. Measurement of Cleaved PARP1 Levels

Analysis of PARP1 degradation (Figure 11) in cancer cells exposed to compounds **I** and **II** (at IC_50_ concentration for each individual line) showed that these compounds caused PARP1 degradation and inhibited its activity. It is worth noting that these in vitro results are a good confirmation of the docking analysis of spiropyrazoline analogues to PARP1 carried out in our previous work. Compound **I** was characterized by a slightly stronger ability to degrade PARP1 in all cancer cells used in the experiment. Clear differences in PARP1 degradation activity between compounds **I** and **II** were observed for NALM-6. For compound **II**, no statistically significant changes in polymerase degradation were observed, while for compound **I**, there was greater degradation of PARP1 compared to the control. A twofold increase in PARP1 degradation with respect to the control was also observed for the HL60 and Ishikawa lines. For the remaining cancer lines for compound **I** and **II**, approximately 50% and 25% increases in polymerase degradation were observed, respectively.

#### 2.3.5. Changes in the Transmembrane Mitochondrial Potential (ΔΨm)

The disruption of the mitochondrial membrane potential (MMP) is one of the first events to occur in a cell following cell damage and induction of apoptosis. MMP changes were tested using the cationic carbocyanine dye JC-1, which is the most commonly used fluorescent probe in the study of mitochondrial membrane potential changes and the detection of its depolarization/hyperpolarization in cells in vitro. The selective accumulation of JC-1 in mitochondria is highly dependent on the transmembrane mitochondrial potential (ΔΨm) [29]. The proapoptotic properties of compound **I** and **II** were confirmed in the analysis of the mitochondrial potential of neoplastic cells treated with the above-mentioned molecules. Both complexes **I** and **II** caused a decrease in the transmembrane mitochondrial potential and a strong depolarization of the mitochondrial membrane of cancer cells. Only in the case of the WM115 line, no statistically significant changes in mitochondrial potential were observed. The greatest changes were observed in HCC38 cells treated with complex **II**—about 60% depolarization of the mitochondrial membranes compared to the control. Slightly smaller changes in mitochondrial membrane depolarization of mitochondrial membranes (~35–40%) were observed for the HL60 and NALM-6 lines, and it is worth noting that compound **I** showed a slightly stronger proapoptotic effect. In the case of other cancer cell lines, a decrease in the transmembrane mitochondrial potential of 15–20% was observed. The changes in mitochondrial potential for cell lines using a JC-1 fluorescent probe are shown in the Figure 12.

## 3. Materials and Methods

### 3.1. Refinement of X-ray Data

Crystals suitable for X-ray measurements were obtained from crystallization in ethanol/methanol solutions after slow evaporation. X-ray data were measured on an Agilent SuperNova Dual Source diffractometer with an Atlas detector using MoKα radiation. All H atoms were fixed geometrically at calculated positions using a riding model. Table 4 reports the results of the crystal structure determinations. Further crystallographic details for the structures reported in this paper may be obtained free of charge upon application to CCDC, 12 Union Road, Cambridge CG21, EZ, UK (fax: (44) 1223-336-033; e-mail: deposit@ccdc.cam.ac.uk), quoting the depository number CCDC 2168428 for compound **I** and CCDC 2168429 for compound **II**.

### 3.2. Molecular Docking Calculation

#### 3.2.1. Ligand Preparation

Prior to docking procedures, the potential inhibitors were prepared using Portable HyperChem 8.0.7 Release (Hypercube, Inc., Gainesville, FL, USA). Each ligand was optimized using a MM + force field and the Polak–Ribiere conjugate gradient algorithm (terminating at a gradient of 0.05 kcal mol^−1^ Å^−1^).

#### 3.2.2. Protein Preparation

The X-ray structures of the PARP1 enzyme used for molecular modeling studies was taken from the Protein Databank (Protein Data Bank accession code: 4HHY). The docking analysis was carried out after standard preparation procedures including:Removal of B, C, and D chains of initial receptor structure;Removal of co-crystalized ligands (including co-crystalized inhibitor);Removal of water molecules;Addition of hydrogen atoms and Gasteiger charges to each atom.

#### 3.2.3. Molecular Docking

Docking of the optimized ligands to the prepared PARP1 enzyme structure was carried out with Autodock Vina 1.1.2 software (The Molecular Graphic Laboratory, The Scripps Research Institute, La Jolla, CA, USA) [35] with exhaustiveness, num modes, and energy_range parameters set as 8, 30, and 10, respectively. For all of the docking studies, a grid box size of 20 Å × 20 Å × 20 Å, centered on the Cα atom of Ser243 (x = −42.067, y = 9.162, z = −14.675) was used. Ser243 was previously mentioned as one of the amino acid residues forming the binding site of PARP1 enzymes [36,37]. The atoms of the protein were fixed during the docking calculations. Graphic visualization of the 3D model for the pose of compound I with the lowest free binding energy of binding was generated using VMD 1.9 software (University of Illinois at Urbana—Champaign, Urbana, IL, USA).

### 3.3. Cell Lines and Cell Culture

Eight human cancer cell lines were used in the experiments—seven adherent human cell lines: Caucasian colon adenocarcinoma COLO205 (ATCC^®^ CCL-222™), endometrium adenocarcinoma HEC-1-A (ATCC^®^ HTB-112™), Ishikawa (99040201; Sigma-Aldrich Sigma, St. Louis, MO, USA), breast epithelial adenocarcinoma cells positive for estrogen and progesterone receptors MCF7 (ATCC^®^ HTB-22™) and negative for expression of estrogen receptor HCC38 (ATCC^®^ CRL-2314™), acute promyelocytic leukemia HL60 (ATCC^®^ CCL-240™), skin melanoma WM115 (ATCC^®^ CRL-1676™), and one suspension cell line, acute lymphoblastic leukemia NALM-6 (purchased from the German Collection of Microorganisms and Cell Cultures). NALM-6, HCC38, and HL60 cells were cultured in RPMI 1640 medium (Thermo Fisher Scientific, Waltham, MA, USA) supplemented with 10% fetal bovine serum and an antibiotic (gentamicin 25 μg/mL) (Gibco, Grand Island, New York, NY, USA). For the remaining tumor cell lines, Dulbecco’s minimal essential medium (DMEM, Lonza, Visp, Switzerland) supplemented with 10% fetal bovine serum and an antibiotic instead of RPMI 1640 was used. Cells were grown at 37 °C in a humidified atmosphere of 5% CO_2_ in air. Exponential growth of cells was maintained by their regular passaging at 90% confluence three times a week using 0.025% trypsin/EDTA (Gibco, Grand Island, New York, NY, USA).

### 3.4. Cytotoxicity Assay

The cytotoxicity was tested by a standard MTT (3-(4,5-dimethylthiazol-2-yl)-2,5-diphenyltetrazolium bromide) (Sigma, St. Louis, MO, USA) microplate cell viability assay [38]. The anticancer activity of the investigated compounds was evaluated in vitro on the basis of their ability to inhibit the proliferation of various cancer cells. Exponentially growing cells were seeded a day before the experiment in a 96-well microplate (Nunc, Roskilde, Denmark) at a density up to 6–8 × 10^3^ cells/mL (depending on the cell line), and exposed to investigated compounds at a wide range concentrations for 48 h. Subsequently, various concentrations (final concentration of 10^−7^–10^−5^ M) of the compounds studied, freshly prepared in DMSO (final concentration <0.1% DMSO) and diluted with complete culture medium, were added. After 48 h of incubation, cells were immediately analysed for the cell viability assay. At this time point, the number of viable cells in each well was estimated by the MTT test. To each microplate well was added 50 µL (5 mg/mL final concentration) of MTT, followed by 3–4 h incubation at 37 °C. After incubation, MTT solution was replaced with 100 µL DMSO/well, which solubilized formazan crystals formed by the action of dehydrogenases of metabolically active cells. Purple formazan absorbance was measured spectrophotometrically at 570 nm. The cytotoxicity of ferrocenes was evaluated on the basis of their IC_50_ concentrations that reduced cell viability by 50% compared to that of untreated control cells, which were arbitrarily taken as 100%.

### 3.5. Measurement of the Changes in Mitochondrial Potential (ΔΨm) Using the Microplate Spectrofluorimetric Method with the JC-1 Fluorescent Probe

The mitochondrial transmembrane potential (MMP, ΔΨm) is generated by an electrical potential created across the inner mitochondrial membrane. Measurement of MMP is useful for assessing mitochondrial function and dysfunction of the mitochondrial membrane; for example, its depolarization and hyperpolarization. The decrease in MMP may be related, inter alia, to the induction of the internal (mitochondrial) apoptotic pathway. Depolarization of the mitochondrial membrane increases the permeability of the mitochondrial membrane, which can lead to the release of apoptosis initiators, such as cytochrome C, and the activation of the apoptotic cascade. JC-1 is a cationic carbocyanine dye widely used as a fluorescent probe to detect mitochondrial membrane depolarization. The selective accumulation of JC-1 in mitochondria is highly dependent on the transmembrane mitochondrial potential. JC-1 enters the negatively charged mitochondrial matrix as a monomer, where it forms fluorescent aggregates with red-orange fluorescence (λ_em_ = 590 nm). During the depolarization of the mitochondrial membrane, or in apoptotic cells, JC-1 remains as a monomer that exhibits green fluorescence (λ_em_ = 525 nm) [39,40].

Cells (10 × 10^3^/well) were seeded in black 96-well plates and grown for 24 h to allow them to reach log phase. After 24 h, the test compounds (IC_50_ concentration) were added and the cells were incubated for 24 h in a CO_2_ incubator at 37 °C. After removing the medium, the cell monolayer was washed twice with PBS and 50 µL of the JC-1 (Sigma, St. Louis, MO, USA) probe at a final concentration of 5 µM was added to the wells of the microplate. Cells were incubated with the dye for 30 min in a CO_2_ incubator. Then, changes in the fluorescence of monomers (λ_ex_ = 485 nm and λ_em_ = 538 nm) and aggregates (λ_ex_ = 530 nm and λ_em_ = 590 nm) of JC-1 were monitored. The results of the tested trials are presented as the ratio of fluorescence of aggregates and JC-1 monomers as a percentage in relation to the ratio of fluorescence of aggregates and JC-1 monomers of the control, which were assumed to be 100%. Cells incubated with 5 µmol of CCCP (carbonyl cyanide 3-chlorophenylhydrazone; Sigma, St. Louis, MO, USA), an organic chemical compound and an ionophore that carries ions across the inner membrane of the mitochondrion, were used as the positive control. CCCP inhibits oxidative phosphorylation by decoupling proton gradients, thereby lowering the electrochemical potential of the inner mitochondrial membrane.

### 3.6. Determination of Apoptotic and Necrotic Cell Fractions by Fluorescence Microscopy (Double Staining of Cells with Fluorescent Dyes Hoechst 33258 and Propidium Iodide)

The simultaneous use of two fluorescent dyes (propidium iodide and Hoechst 33258; Sigma, St. Louis, MO, USA), whose mechanism of penetration into the cell and the fluorescence spectrum are different, allows the identification of four types of cells in the same sample: live, early apoptotic, late apoptotic, and necrotic. Propidium iodide (PI) has a negative charge and only penetrates cells with damaged cell membranes, which allows the identification of necrotic cells or those present in late phases of apoptosis. Hoechst 33258 freely penetrates through the intact membrane of living and early apoptotic cells, thus enabling their identification. As a result of the dye penetration through intact biological membranes, it stains the DNA of the cell nucleus a light blue color.

Both fluorochromes are excited by ultraviolet light—propidium iodide has orange-red fluorescence, while Hoechst 33258 has blue fluorescence. Staining of cells with a mixture of both dyes allows one to distinguish four fractions of cells differing in fluorescence in the microscopic image:Live cells (weak, dull, light blue fluorescence);Cells in the early phase of PCD (bright, light blue fluorescence);Cells in the late phase of PCD (pink-purple fluorescence);Necrotic cells (intense red fluorescence).

Cells were seeded in 12-well plates in appropriate culture medium 24 h prior to experimentation. After this time, the analyzed compounds were added to the cells at the determined IC_50_ concentration. Cells were incubated with compounds for 24 h. At the end of incubation, the medium from each well was removed and 3 mL of HBSS containing Hoechst 33258 (0.13 mM) and PI (0.23 mM) were added. The cells were incubated with the fluorochromes for 5 min at room temperature, in the dark. After this time, HBSS with fluorochromes was removed and fresh HBSS was added to the cells. The analysis was performed with a fluorescence microscope (Olympus IX70, Japan) under 400× magnification. Cells were classified as live, apoptotic, or necrotic on the basis of their morphological and staining characteristics.

### 3.7. Analysis of DNA Damage Using the Alkaline Version of the Comet Assay (Single Cell Electrophoresis)

Genetic damage resulting from cell exposure to genotoxic agents leads to structural disorders of DNA and is a potential source of mutation. Single cell electrophoresis allows, among other functions, the measurement of single-stranded and double-stranded DNA breaks. During electrophoresis, DNA migrates towards the anode at a rate dependent on the size of the molecule. The microscopic image of the damaged cell subjected to electrophoresis resembles a comet: the “head” corresponds to the place where the cell was immobilized before lysis, the “tail” consists of loops and fragments of DNA strands relaxed and released from the nuclear structures as a result of DNA breaks [41].

Cells (1 × 10^6^) were seeded in 35 mm dishes and grown in medium for 24 h to allow them to reach log-phase growth. After 24 h, the tested compounds (IC_50_ concentration) were added to the cell culture medium. Cells were incubated with compounds for 24 h. After incubation, the medium was replaced with a new one, and a further 24 h post-incubation culture was performed to allow cells to repair DNA damage. Immediately prior to analysis, cells were trypsinized, centrifuged, resuspended in a small amount of PBS, and added to a low-melting-point agarose solution. A small amount of the agarose suspension of cells prepared in this way was placed on a glass slide, coated with normal-melting-point agarose, covered with a coverslip, and left until the agarose solidified. Slides were incubated for at least 1 h in lysis buffer (2.5 M NaCl, 100 mM, Na2-EDTA, 10 mM, Tris, 1% Triton X-100) to release DNA. The slides were then washed three times with development buffer (1 mM, Na2-EDTA, 300 mM NaOH) and incubated in the same buffer for 20 min. The next step was electrophoresis of cells in electrophoretic buffer for 20 min (29 V, 30 mA). After electrophoresis, the slides were allowed to dry, and were then stained with DAPI solution (2 mg/mL). The comet DNA analysis was performed under a Nikon Eclipse fluorescence microscope (Nikon, Tokyo, Japan) equipped with a 4910 COHU video camera (Cohu, Inc., San Diego, CA, USA) and a computer with Lucia-Comet v. 4.51 software installed (Imaging Laboratory, Prague, Czech Republic). Each time, from each preparation, 100 randomly selected comets were counted.

### 3.8. Measurement of Cleaved PARP Levels

Cells (1 × 10^6^) were seeded in 6-well plates in appropriate culture medium 24 h prior to experimentation. After this time, the analyzed compounds were added to the cells at the determined IC_50_ concentration. Cells were incubated with compounds for 24 h. At the end of incubation, cells were washed in ice-cold PBS, and lysed in RIPA buffer (50 mM Tris-HCL, pH 8.0, with 150 mM sodium chloride, 1.0% Igepal CA-630 (NP-40), 0.5% sodium deoxychlorate, and 0.1% sodium dodecyl sulfate) with protease inhibitor (phenylmethylsulfonyl fluoride) (Sigma-Aldrich, Sigma, St. Louis, MO, USA). The level of cleaved PARP was estimated using the PARP Cleaved [214/215] ELISA kit (Invitrogen, CA, USA) according to the protocol described in manufacturer’s instructions.

### 3.9. Statistical Analysis

All data are expressed as the mean ± SEM and presented as a percentage of control (untreated) cells, which were taken as 100%. Normality of data was tested with the Shapiro–Wilk test, and homogeneity of variance was verified with Levene’s test. The significance of the differences between pairs of means was estimated using one-way ANOVA and post hoc Tukey’s test. All statistics were calculated with the STATISTICA statistical software package (StatSoft, Tulsa, OK, USA). A * *p* value of <0.05 was considered significant.

### 3.10. Chemistry

Synthesis of compound **I** and compound **II** was described in the previous paper [17] based on the article by Pijewska [42]. Both compounds were purified by crystallization from methanol. The structures of compound **I** and **II** were characterized using IR, ^1^*H*- and ^13^*C*-NMR, MS spectroscopy, and elemental analysis. All solvents used in the synthesis were purchased from Sigma-Aldrich (St. Louis, MO, USA) and were used without further purification. The infrared transmission spectra of the crystalline products were recorded using a Nexus Thermo Nicolet FT-IR spectrophotometer (Wien, Austria; Faculty of Chemistry, University of Lodz). The MS-ESI were measured at the University of Lodz Faculty of Chemistry on a 500–MS LC Ion Trap mass spectrometer (Varian, Palo Alto, CA, USA). Elemental analyses were performed in the Faculty of Chemistry (University of Lodz) using a Vario Micro Cube (Langenselbold, Germany) by Elemental analyzer. ^1^*H*- (600 MHz) and ^13^*C*-NMR spectra (150 MHz) were recorded at the University of Lodz Faculty of Chemistry on a Bruker Avance III instrument (Bruker, Billerica, MA, USA). The samples of compound **I** and **II** were dissolved in deuterated DMSO. The chemical shifts are given in ppm, and the coupling constants in Hz. Melting points were determined on a B-540 Melting Point apparatus (Büchi, Flawil, Switzerland) in capillary mode, and they were uncorrected.

#### 3.10.1. Synthesis of 5′-(3-Methylphenyl)-2-phenyl-4′,5′-dihydro-4H-spiro[chromano-3,3′-pirazol]-4-one (Compound **I**)

A detailed description of the physicochemical properties and spectra of the tested compounds is provided in the Appendix A.

#### 3.10.2. Synthesis of 5′-(4-Methylphenyl)-2-phenyl-4′,5′-dihydro-4H-spiro[chromano-3,3′-pirazol]-4-one (Compound **II**)

A detailed description of the physicochemical properties and spectra of the tested compounds is provided in the Appendix A.

#### 3.10.3. Spectroscopy UV–VIS

For both compounds, UV spectral studies were performed. Ethanol was used as a solvent. The maximum absorption for compound **I** was determined to 268.8 nm, and for compound **II** to 261.2 nm (Figure 13). The extinction coefficients were determined in the concentration range of 6 × 10^−5^ M–1 × 10^−5^ M. They were respectively 13,367 dm^3^cm^−1^mol^−1^ (compound **I**) (Figure 14) and 13,358 dm^3^cm^−1^mol^−1^ (compound **II**) (Figure 15), with regression coefficients (r^2^) of 0.9999.

## 4. Conclusions

Our current work presents studies of the biochemical and crystal structure, and docking, of spiropyrazoline compounds. To check the potential binding properties of both synthesized compounds to PARP1, the docking simulations were performed. The obtained results indicate that compound **I** demonstrated a slightly lower free energy of the binding value (−9.7 kcal mol^−1^), which indicates that the presence of the methyl group at the *meta* position may be more favorable in binding with the active site. It was noticed that the methyl group of compound **I** is close distances to the Ile214 and Leu216 residues, suggesting the presence of hydrophobic interactions, which can theoretically stabilize the ligand–enzyme complex. Furthermore, other plausible interactions (e.g., π–π interactions with the Tyr235 and Tyr246 residues) were also detected. In addition to in silico studies, we conducted an in vitro evaluation of the basic molecular mechanisms of the anticancer activity of the compounds tested. The cytotoxicity assessment showed that the compounds have extremely promising antiproliferative properties in relation to the eight cancer cell lines used. Basic studies with the use of fluorescent probes provide evidence that one of the mechanisms of biological activity of the analyzed compounds is the activation of apoptosis of cancer cells. In addition, the initiation of programmed death of tumor cells was associated with the high genotoxicity of compound **I** and **II**, which was observed in all cancer cell lines used. Inhibition of DNA repair in cancer cells exposed to the test compounds may result from their interaction with the PARP1 protein and its degradation. In silico studies of the molecular docking of the analyzed compounds to PARP1 were confirmed in vitro using the ELISA assay. A clear increase in degraded PARP1 was observed in cells treated with both compound **I** and **II**. These results constitute an extremely valuable basis for further research on possible molecular mechanisms of the antitumor activity of the analyzed compounds, the more so as PARP1 is currently one of the main goals of modern anticancer therapy. It should be emphasized that, on the basis of the analysis of cytotoxic properties, it is difficult to unequivocally state which compound is characterized by better antiproliferative activity. The activity of a compound depends not only on its structure, but also on the type of tumor and its properties. It is worth emphasizing, for example, two lines of breast cancer—HCC38 which is negative for expression of estrogen receptor (ER) and MCF7 with estrogen glucocorticoid receptors. Both tested compounds were more active towards MCF7, which may indicate the interaction of the compounds with estrogen receptors and thus an increase in their biological antitumor activity. Moreover, it is worth noting that compound **I** showed a better degradation activity of the enzyme involved in DNA repair—PARP1. In vitro studies confirm molecular docking to the enzyme, where compound **I** has been shown to have a higher PARP1 inhibitory activity. The crucial element seems to be the attachment of a methyl group in the meta position to the terminal aromatic ring connected to the pyrazolidine unit. Studies on PARP1 degradation are complemented by the genotoxicity analysis of spiropyrazoline derivatives. Again, the comet test confirmed the superior biological activity of compound **I** in terms of DNA damage. It is therefore likely that this compound interacts primarily with DNA, causing severe damage; what is more, it blocks the repair of DNA strands in cancer cells, e.g., by degrading PARP1. On the other hand, compound **II** led to a slightly stronger depolarization of the mitochondrial membranes of tumor cells, causing their apoptosis. It is therefore likely that the biological activity of derivative **II** may be carried out in other biological pathways where mechanisms other than genotoxicity are involved.

## Figures and Tables

**Figure 1 ijms-23-06061-f001:**
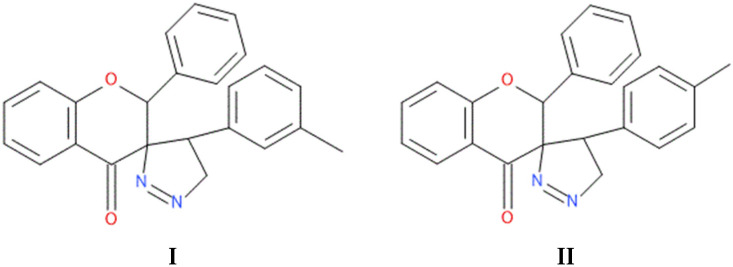
The chemical structure of compound **I** and **II**.

**Figure 2 ijms-23-06061-f002:**
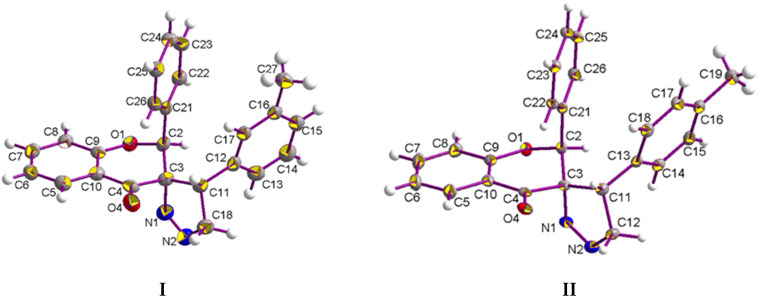
The molecular structure of compound **I** and **II** with displacement ellipsoids drawn at the 50% probability level.

**Figure 3 ijms-23-06061-f003:**
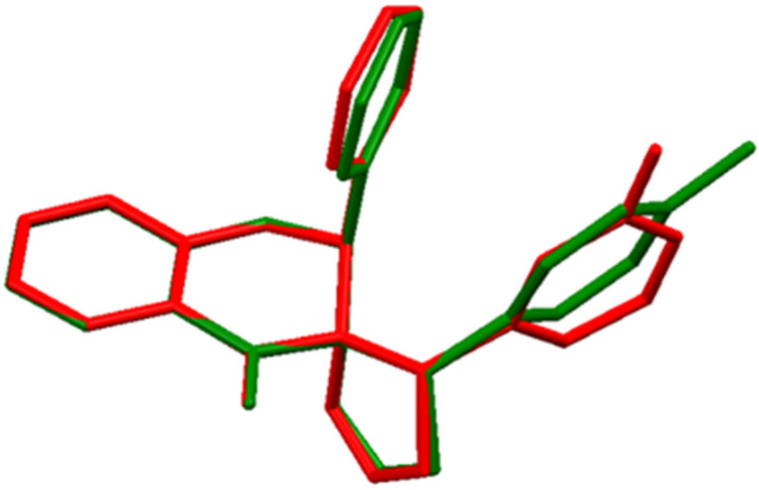
Molecular overlays of two derivatives: compound **I**—red, compound **II**—green. Hydrogen bonds are omitted for clarity.

**Figure 4 ijms-23-06061-f004:**
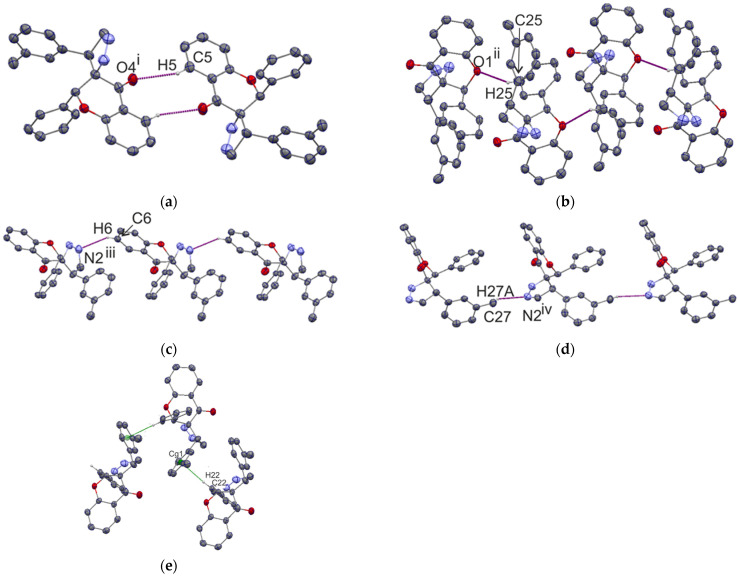
Fragment of the crystal packing of compound **I**, showing the formation of the R_2_^2^(10) ring (**a**), C7 chain along [010] (**b**), C8 chains along [100] and [001] (**c**,**d**), and C–H…π interaction (**e**).

**Figure 5 ijms-23-06061-f005:**
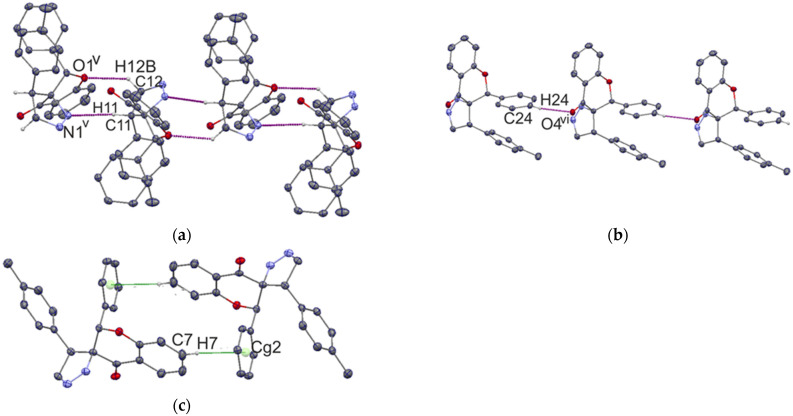
Fragment of the crystal packing of compound **II**, showing ring R_2_^2^(8) formed on the base of two chains: C(6) and C(4) formed by C12–H12B…O1^v^ and C11–H11…N1^v^ hydrogen bonds (**a**), and chain C(8) formed by a C24–H24…O4^vi^ hydrogen bond (**b**). The dimer formed by C–H…π interactions is also presented (**c**).

**Figure 6 ijms-23-06061-f006:**
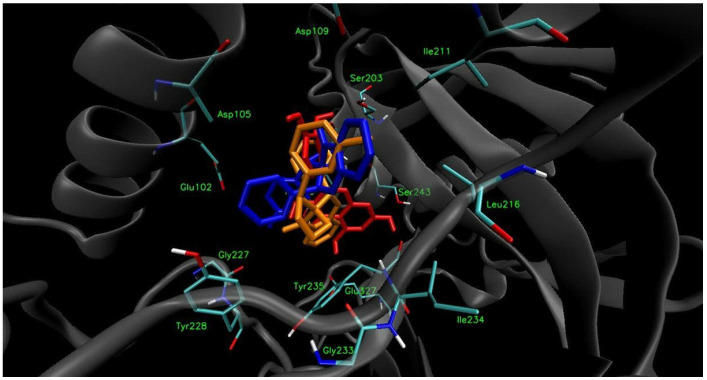
Docked binding modes of compound **I** (orange) and **II** (blue), and quercetin (red) to the PARP1 enzyme’s active site.

**Figure 7 ijms-23-06061-f007:**
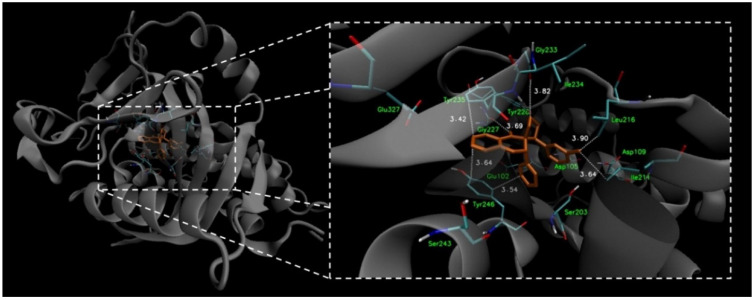
Docked binding mode and distances to the amino acid residues of PARP1 enzyme’s active site for compound **I**.

**Figure 8 ijms-23-06061-f008:**
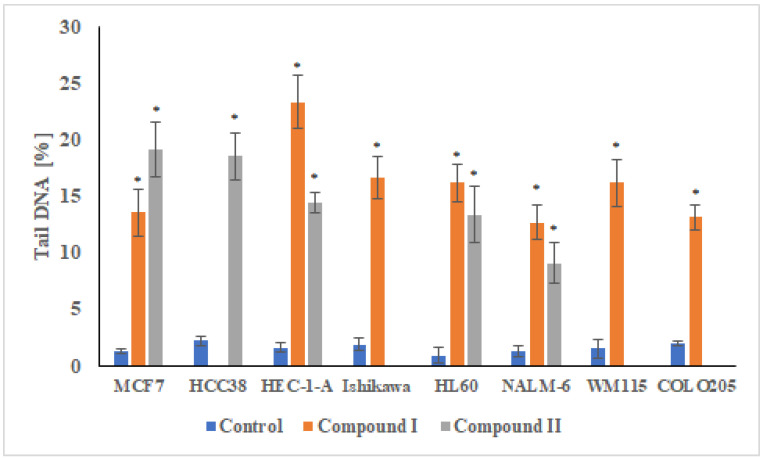
DNA content in the tail of the comet obtained from MCF7, HCC38, HEC-1-A, Ishikawa, COLO205, HL60, NALM-6, and WM115 treated with control and test compounds for 24 h. The DNA tail fraction was estimated at the end of the incubation period using an alkaline version of the comet assay. Each experiment was performed at least in triplicate. The results represent mean ± SEM of the data from 3 individual experiments, and for each analysis, 100 randomly chosen comets were counted, * *p* < 0.05 vs. control.

**Figure 9 ijms-23-06061-f009:**
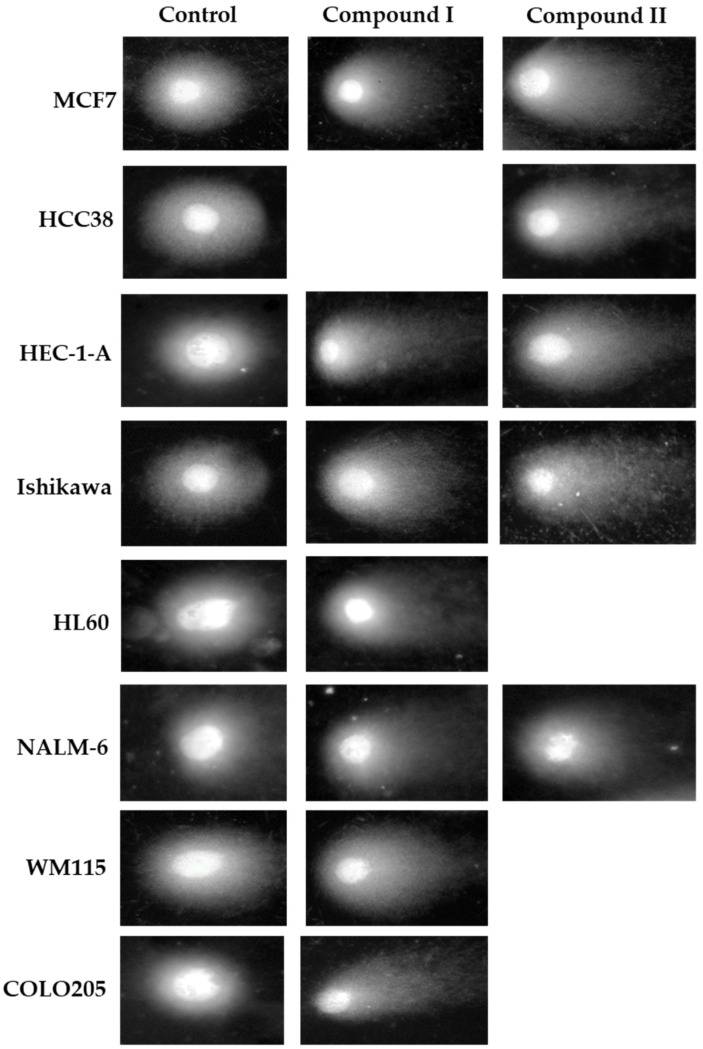
DNA comets obtained from MCF7, HCC38, HEC-1-A, Ishikawa, COLO205, HL60, NALM-6, and WM115. Cells were incubated with control and test compounds for 24 h at concentrations of the calculated IC_50_.

**Figure 10 ijms-23-06061-f010:**
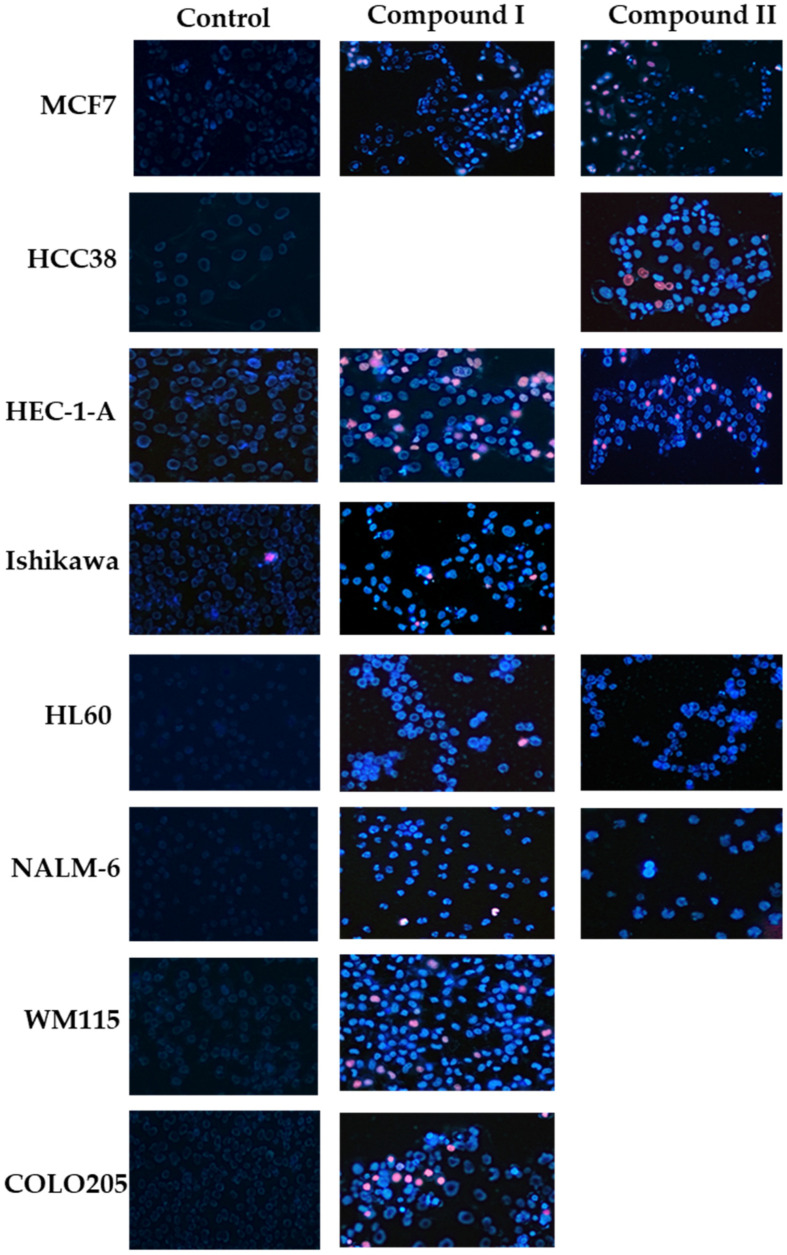
Analysis of the physiological state of single cancer cells MCF7, HCC38, HEC-1-A, Ishikawa, COLO205, HL60, NALM-6, and WM115 by fluorescent staining with the use of two fluorescent dyes—Hoechst 33258 and propidium iodide. Cancer cells were stained with fluorescent dyes to assess the integrity of the cell membrane and determine the fraction of living cells (pale blue fluorescence) and dead cells—including early/late apoptosis (bright blue/violet fluorescence) and necrosis (red fluorescence). Cells were incubated with derivatives for 24 h at concentrations of the calculated IC_50_.

**Figure 11 ijms-23-06061-f011:**
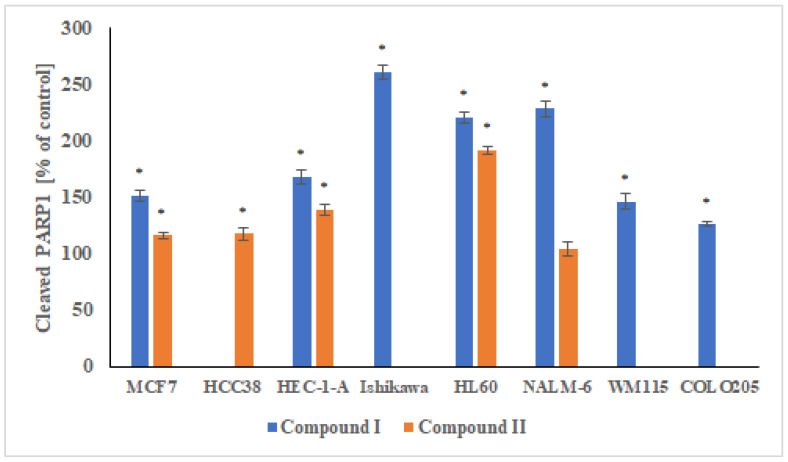
Measurement of PARP1 degradation in MCF7, HCC38, HEC-1-A, Ishikawa, COLO205, HL60, NALM-6, and WM115 cells exposed to compound **I** and **II**. Cells were incubated with derivatives for 24 h at concentrations of the calculated IC_50_. The results represent mean ± SEM of the data from 3 individual experiments, untreated control cells arbitrarily taken as 100%, * *p* < 0.05 vs. control.

**Figure 12 ijms-23-06061-f012:**
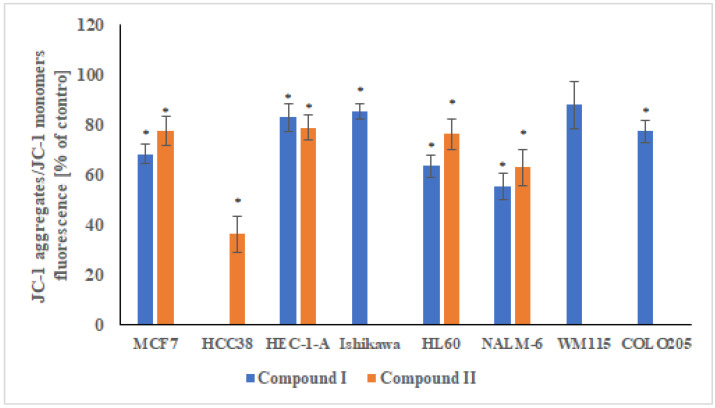
Changes in the mitochondrial potential of MCF7, HCC38, HEC-1-A, Ishikawa, COLO205, HL60, NALM-6, and WM115 cells exposed to compound **I** and **II.** Cells were incubated with derivatives for 24 h at concentrations of the calculated IC_50_. The results represent mean ± SEM of the data from 3 individual experiments, untreated control cells arbitrarily taken as 100%, * *p* < 0.05 vs. control.

**Figure 13 ijms-23-06061-f013:**
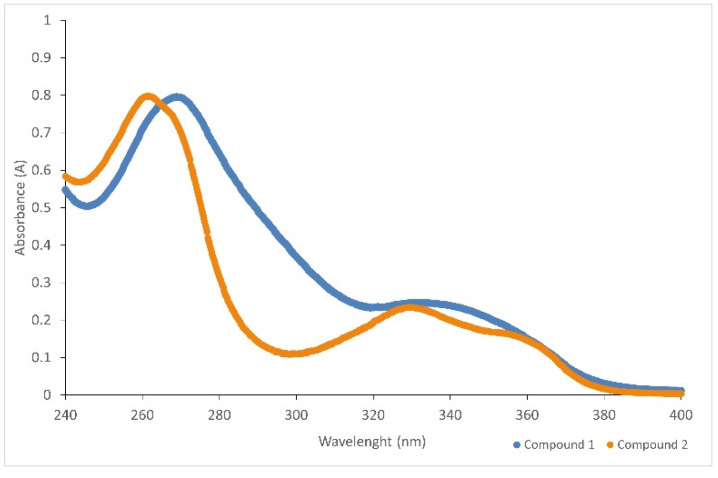
The absorption spectrum of compound **I** and **II**.

**Figure 14 ijms-23-06061-f014:**
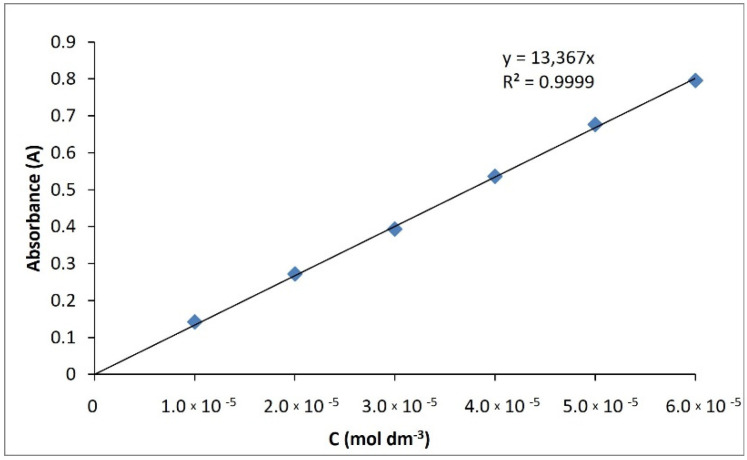
Calibration curve of compound **I**.

**Figure 15 ijms-23-06061-f015:**
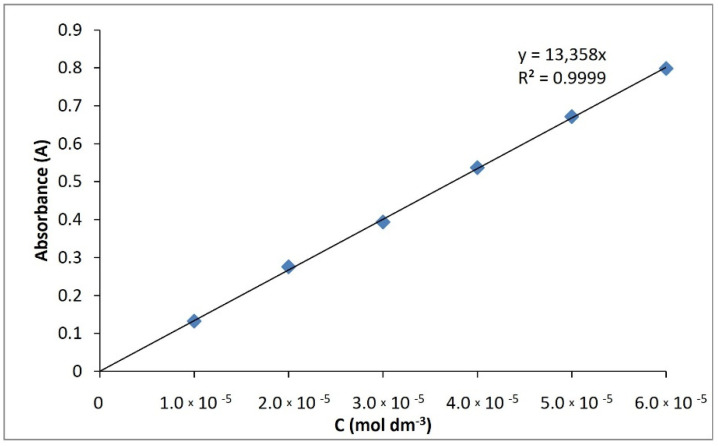
Calibration curve of compound **II**.

**Table 1 ijms-23-06061-t001:** Hydrogen bond geometry (Å, °) for compound **I** and **II**.

D–H···A	D–H	H···A	D···A	D–H···A
**Compound I**				
C5–H5···O4^i^	0.95	2.51	3.235(3)	133
C25–H25···O1^ii^	0.95	2.69	3.587(3)	159
C6–H6···N2^iii^	0.95	2.66	3.308(4)	126
C27–H27A···N2^iv^	0.98	2.71	3.483(4)	136
**Compound II**				
C12–H12B··O1^v^	0.99	2.46	3.224(2)	134
C24–H24···O4^vi^	0.95	2.43	3.170(2)	135
C11–H11···N1^v^	1.00	2.58	3.575(2)	172

Symmetry codes: (i) −x, −y, −z; (ii) −x, y − 1/2, −z + 1/2; (iii) x − 1, y, z; (iv) x, −y + 1/2, z + 1/2; (v) 1/2 − x, −1/2 + y, 1/2 − z; (vi) 1 + x, y, z.

**Table 2 ijms-23-06061-t002:** Free energies of binding calculated for compounds **I** and **II**, and quercetin.

No	Structure	Free Energy of Binding (kcal mol^−1^)
Compound **I**	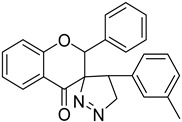	−9.7
Compound **II**	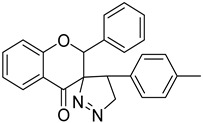	−8.7
Quercetin	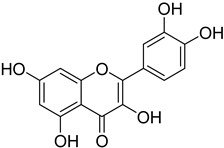	−8.9

**Table 3 ijms-23-06061-t003:** IC_50_ concentrations, in the µM range, of compound **I** and **II** against MCF7, HCC38, Ishikawa, and HEC-1-A cells. The cells were incubated with the compounds for 24 h (cytotoxic effect); results are expressed as means ± SEM of 3 individual repeated experiments.

Number of Compound	IC_50_ Concentration (µM)			
	MCF7	HCC38	Ishikawa	HEC-1-A
**Compound I**	43.1 ± 1.5	57.3 ± 1.1	34.2 ± 1.7	39.3 ± 1.3
**Compound II**	31.4 ± 0.9	37.2 ± 1.9	56.7 ± 2.2	27.4 ± 0.8
**4-Chromanone ***	>500	>500	>500	>500
**Cisplatin ***	11.4 ± 1.3	29.6 ± 2.2	16.1 ± 2.5	89.5 ± 4.7

***** Reference compounds.

**Table 4 ijms-23-06061-t004:** Experimental details.

	(Compound I)	(Compound II)
**Crystal data**		
Chemical formula	C_24_H_20_N_2_O_2_	C_24_H_20_N_2_O_2_
M_r_	368.44	368.42
Crystal system, space group	Monoclinic, P2_1_/c	Monoclinic, P2_1_/c
Temperature (K)	100	100
a, b, c (Å)	9.2373 (5), 9.7336 (5), 20.4183 (17)	9.2071 (2), 10.5662 (2), 19.6790 (4)
Β (°)	90.641 (6)	93.778 (2)
V (Å3)	1835.7 (2)	1910.29 (7)
Z	4	4
Radiation type	Mo Kα	Mo Kα
Μ (mm^−1^)	0.09	0.08
Crystal size (mm)	0.4 × 0.05 × 0.03	0.32 × 0.03 × 0.02
**Data collection**		
Diffractometer	SuperNova, Dual, Cu at zero, Atlas	SuperNova, Dual, Cu at zero, Atlas
Absorption correction	*XABS*2 [30]	Multi-scan *CrysAlis PRO* 1.171.38.41q [24] Empirical absorption correction using spherical harmonics, implemented in SCALE3 ABSPACK scaling algorithm.
T_min_, T_max_	0.484, 0.979	0.808, 1.000
No. of measured independent and observed [*I* > 2σ(*I*)] reflections	3803, 3803, 2670	15831, 3957, 3351
*R* _int_	0.070	0.029
(sin θ/λ)_max_ (Å^−1^)	0.628	0.628
**Refinement**		
*R*[*F*^2^ > 2σ(*F*^2^)], *wR*(*F*^2^), *S*	0.068, 0.200, 1.03	0.037, 0.100, 1.04
No. of reflections	3803	3957
No. of parameters	254	254
No. of restraints	1	0
H atom treatment	H atom parameters constrained	H atom parameters constrained
Δρ_max_, Δρ_min_ (e Å^−3^)	0.60, −0.39	0.25, −0.26

Computer programs: CrysAlis PRO 1.171.38.41q [31], SHELXT [32], SHELXL [33], DIAMOND [34].

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
