# Peer review of "Biochemical, Structural Analysis, and Docking Studies of Spiropyrazoline Derivatives"

_ijms, 2022, doi:10.3390/ijms23116061_

Round 1

Reviewer 1 Report

Abstract is very unclearly written, looks more like a summary instead of laying out the fundamental research idea. 
Introduction on PARP is missing references (e.g. claim of proliferation, apoptosis regulation etc.)

The authors recite their entire paper from reference 1 in the introduction. This is unnecessary and not useful for this study.

In table 2, the structures have been ruined by formatting.

Figure 6: I would like to see a closer image from the docked pose for both compounds, preferentially overlaid with a known inhibitor. On that note: it needs to be explained exactly how the structure was prepared for docking and why the site was chosen. Was the inhibitor in the PDB file removed before docking? Was the protein kept flexible?

Results like the cytotox assay, comet assay and so on need a brief description on what is being done. Additionally, instead of showing bar graphs, full signal vs concentration graphs would be much more informative instead of doing a lot of different cell lines. I would focus more on full graphs of all tested concentrations for:
cytotoxicity
PARP1 degradation
changes in mitochondrial potential

In the comet assay: how is the DNA quantified to percentages?

Would like to see the apoptotic effect quantified in a way. cell counting tools could help here.

2.3.4.: The first paragraph belongs in the introduction section

From here on out, some cell lines only contain either compound I or II. This is very confusing. Would focus on cell lines that contain both compounds. 

Overall, more emphasis needs to be out on the differences and discussion of those differences between to two compounds.

Synthesis and identification spectra in supplementary are missing 

Author Response

Response to Referee

We thank the referee for positive comments and thoughtful review of the manuscript. We have made revisions according to your comments and suggestions, as described below. Changes made to the manuscript are marked in blue.

Reviewer #1:

Comment 1. Abstract is very unclearly written, looks more like a summary instead of laying out the fundamental research idea. Introduction on PARP is missing references (e.g. claim of proliferation, apoptosis regulation etc.)

Response 1

According to the suggestion, the abstract was rewritten. The corrected abstract is presented below:

In this study we evaluated the antiproliferative potential, DNA damage, crystal structures and docking calculation of two spiropyrazoline derivatives. The main focus of the research was to evaluate the antiproliferative potential of synthesized compounds towards 8 cancer cell lines. Compound I demonstrated promising antiproliferative properties, especially toward the HL-60 cell line, which IC50 is equal to 9.4 µM/L respectively. The analysis of DNA damage by the comet assay showed that compound II caused DNA damage to tumor lineage cells to a greater extent than compound I. The level of damage to tumor cells of the HEC-1-A lineage was 23%. The determination of apoptotic and necrotic cell fractions by fluorescence microscopy indicated that cells treated with spiropyrazoline-based analogues were entering the early phase of programmed cell death. Compounds I and II depolarized the mitochondrial membranes of cancer cells. Furthermore, we performed the simple docking calculations, which indicated that the obtained compounds are able to bind to the PARP1 active site, at least theoretically (the free energy of binding values for compound I and II were -9.7, and 8.7 kcal mol-1, respectively). In silico studies of the influence of the studied compounds on PARP1 were confirmed in vitro with the use of 8 cancer cell lines. The degradation of the PARP1 enzyme was observed, with compound I characterized by a higher protein degradation activity.

Comment 2: The authors recite their entire paper from reference 1 in the introduction. This is unnecessary and not useful for this study.

Response 2

The authors thank the Reviewer for paying attention. The authors corrected the introduction. The revised excerpt from the introduction is below:

We have been conducting research on the biological properties of pyrazolines condensed with a chromone ring substituted with various functional groups for a long time [1,2]. It has been confirmed that flavone derivatives exhibit a number of biological properties [3,4], thanks to the presence of heteroatoms in their structure, however, our team pays the greatest attention to anti-cancer properties. Due to the combination of the pyrazoline core containing nitrogen atoms with flavonoid derivatives, the biological potential of the compounds is greater. Over the course of several decades, many scientific papers have been created based on the research carried out on the properties of flavonoid compounds [4–6]. Scientists have paid a lot of attention to anti-cancer and antioxidant activities [7]. Despite many documented scientific reports, further research is being carried out on further biological properties of these compounds.

Comment 3: In table 2, the structures have been ruined by formatting.

Response 3

The formatting is corrected (it depends on the version of Microsoft word editor).

Comment 4: Figure 6: I would like to see a closer image from the docked pose for both compounds, preferentially overlaid with a known inhibitor. On that note: it needs to be explained exactly how the structure was prepared for docking and why the site was chosen. Was the inhibitor in the PDB file removed before docking? Was the protein kept flexible?

Response 4 

According to the reviewer’s suggestion, the image with overlaid compounds I, II and quercetin (used as a reference PARP1 inhibitor) was added.

Regarding the structure preparation, please see the methodology section, where we described all procedures in detail (including the ligands and enzyme preparation). You can also find there information that prior to docking calculations, we removed all ligands from the PDB structure – including inhibitor. We modified slightly this part to provide more details (e.g., we explained why we used Ser243 as a center of the grid box and we added information that the enzyme atoms were fixed during the docking calculations).

Comment 5: Results like the cytotox assay, comet assay and so on need a brief description on what is being done. Additionally, instead of showing bar graphs, full signal vs concentration graphs would be much more informative instead of doing a lot of different cell lines. I would focus more on full graphs of all tested concentrations for: cytotoxicity PARP1 degradation changes in mitochondrial potential

Response 5

Thank you for your attention to the presentation of the results. We tried to describe all experiments as accurately as possible both in terms of preparation and execution (mathematics and methods section), as well as the obtained results and their analysis. Under each graph with the analysis, we briefly describe the obtained results and analyze their significance. We realize that there is a lot of information in the charts that we present, and this is due to the large number of cancer cell lines used. Because we have collected the cytotoxicity results for all the compounds and cancer cell lines used in the table, we decided that it would be appropriate to include the type of cancer line in the graphs. In each experiment, cells were exposed to the compounds at the IC50 concentration calculated for a given line. Therefore, we hope that, on the data from the cytotoxicity table, the reader will be able to read all the graphs in a clear and transparent manner, and, more importantly, compare the biological effect of the analyzed spiropyrazoline derivatives on different cancer cell lines. At the same time, we would like to point out that the reviewer's remark is an extremely valuable hint for the future, especially from the point of view of the characteristics of our research and the way it is presented.

Comment 6: In the comet assay: how is the DNA quantified to percentages?

Response 6

The comet test is the basic method of analyzing the degree of DNA fragmentation caused by genotoxic factors. This method identifies single- and double-stranded DNA breaks as well as any chemical and enzymatic modifications that can turn into DNA breaks or chromatids. The comet test enables the detection of DNA damage at the level of a single cell. Lysed cells are electrophoresed and stained with a fluorescent substance (e.g. DAPI dye). One obtains an image in the form of "comets". The head is where the cell immobilizes before lysis, and the tail is the damaged DNA fragments. The measure of the level of DNA damage is the length of the tail and the amount of DNA it contains. The slides were analyzed under a fluorescence microscope equipped with a filter adapted to the fluorescent dye previously used. Processing of the computer image was performed in our case with Lucia-Comet v. 4.51 program. The DNA damage percentage is not calculated by us but by Lucia-Comet, based on measurements of the color intensity and the length of the comet's tail. This program analyzes the intensity of fluorescence and on this basis calculates a series of parameters characterizing the amount of DNA damage for each cell. The percentage of DNA in the comet's "head" corresponds to the genetic material located in the nucleus, and the fluorescent region of the "tail" of the comet corresponds to the genetic material that has left the cell nucleus as a result of migration in an electric field. The most frequently used parameter for assessing DNA damage is the tail moment, i.e. the product of the percentage of DNA in the comet's "tail" and its length. For a better visualization of the program and the comet test, we include a photo from the Lucia-Comet program.

Comment 7: Would like to see the apoptotic effect quantified in a way. cell counting tools could help here.

Response 7

Obviously, this is a very valuable observation from the point of view of understanding the molecular mechanisms of the anticancer activity of the compounds analyzed. In this study, we focus primarily on the interaction of compounds with the PARP protein and the possible effects of the degradation of this enzyme on cancer cells. In the first stage of research on spiropyrazoline derivatives, we focused on basic research that will answer the fundamental questions - Are the compounds pro-apoptotic or genotoxic. By making a qualitative assessment of the programmed cancer cell death pathway, we wanted to make a preliminary analysis of the possible molecular pathways of the anticancer activity of the derivatives. Because of the very satisfactory cytotoxicity results of spiropyrazoline derivatives and their genotoxic/ apoptotic properties, we will continue our work on their biological activities. In future studies, of course, we plan to significantly expand the scope of research (also on apoptosis), where we will certainly perform a quantitative analysis, among others. using a cytometer.

Comment 8: 2.3.4.: The first paragraph belongs in the introduction section

Response 8:

The first paragraph of subsection 2.3.4. was moved to the introduction.

Comment 9: From here on out, some cell lines only contain either compound I or II. This is very confusing. Would focus on cell lines that contain both compounds. 

Response 9:

In the first stage of biological research on the molecular mechanisms of anticancer activity of spiropyrazoline derivatives, we determined their cytotoxic properties against 8 cancer cell lines. This study was aimed at, inter alia, screening of cytotoxic properties of tested compounds and selection of the most active ones for further stages of research. As the breakpoint, we chose the IC50 =50 µM (noted in the article under section 2.3.1. Cytotoxic activity). In the case when the compound was characterized by a low biological effectiveness (IC50 above 50 µM) and clearly did not inhibit cell profiling of a given tumor line, it was not analyzed for DNA damage, apoptosis or PARP degradation. Hence, following determination of cytotoxicity, some cell lines were tested solely against compound I or II. We tried to select the most attractive cell lines and compounds that would be most attractive in the preliminary research of biological antitumor activity from the point of view of their potential use as chemotherapeutic agents. To increase the transparency of the article, we have added an appropriate sentence explaining which compounds were subjected to a detailed biological analysis against which biological cell lines. Line 244 – “Based on the IC50 cytotoxicity results in the MCF-7, HEC-1-A, HL-60 and NALM-6 lines, both compounds I and II were tested. For the Ishikawa, WM115 and COLO205 lines only the biological activity of compound 1 was analyzed while for HCC38 only compound II.

Comment 10: Overall, more emphasis needs to be out on the differences and discussion of those differences between to two compounds.

Response 10:

Of course, we agree with this valuable remark regarding the need to discuss the differences in the biological activity of the analyzed compounds. Our research is of a basic nature. Through planned experiments, we tried to obtain answers to the questions whether the analyzed derivatives exhibit anti-proliferative activity in relation to selected types of cancer and what are the possible pathways of their anti-cancer mechanisms. We focused primarily on the qualitative assessment of the apoptotic pathway, genotoxic properties and analysis of PARP1 degradation. On the basis of these studies, it is difficult to determine the exact mechanism of the antitumor activity of the analyzed derivatives, whether, for example, apoptosis is carried out by the receptor or mitochondrial pathway, and what is the role of ROS in shaping the general cytotoxicity of compounds. In the future, we want to expand the scope of molecular tests that will answer these and other questions regarding the precise mechanisms of antitumor activity of the spiropyrazoline derivatives under study. However, bearing in mind the reviewer's request, in the summary we made a short analysis of the obtained results and the differences in biological activity of both compounds.

Comment 11: Synthesis and identification spectra in supplementary are missing 

Response 11

The synthesis and NMR and IR spectra were added in supplementary materials.

Sincerely yours,

Prof. Elzbieta Budzisz

Reviewer 2 Report

The manuscript titled “Biochemical, structural analysis and docking studies of spiro-2 pyrazoline derivatives” is concerned with the use of chemically synthesized spiropyrazoline derivatives to inhibit cancer proliferation. The aim of the study was to investigate the biochemical analysis, structural characterization, and docking studies of the analogues with a clear focus on its antiproliferative potential against 8 cancer cell lines. The aim of the study was achieved by Refinement of X-ray data. Molecular docking was studied by Autodock Vina software after receptor and ligand preparation. The cytotoxicity activity of the ligands was confirmed by MTT assay and mitochondrial potential was done by microplate spectrofluorimetric method with the JC-1 fluorescent probe. In addition, apoptotic analysis was evaluated by fluorescence microscopy. DNA damage was determined by comet method.

Both compounds possess antiproliferative activities with low IC50. Furthermore, comet assay showed that compound 2 caused DNA damage when compared to compound 1. Cancer cells dosed with these compounds entered the early phase of programmed cell death but compound 1 showed a better apoptotic activity against the studied cancer cells. Compound I and II depolarized 30 the mitochondrial membranes of cancer cells.

This study is interesting and straight forward. However, some issues need to be addressed. These concerns are listed below.

Major concerns

Results and discussion section

The docking results of the two compounds should be further compared with the co-crystalized ligand of the receptor or the anti-cancer drug stated previously in line 67. Furthermore, the stated energy of the complexes are the static binding energies which is often not too accurate.

I will suggest that authors carry out the dynamic binding energy of the two complexes using MM/GBSA or PB/GBSA to correct any doubt.

Is there any difference with respect to interacting atoms of the receptor between the two compounds when interacting with PARP1?

In the cytotoxicity activity section (Line 205): It is worth noting the selectivity of the compounds between normal and cancer cells. The cytotoxicity of these compounds should be evaluated against the corresponding normal cells of the studied cancer cells.

For the comet assay, the result should be compared to a control.

Minor comments

Abstract section

The abstract feels too lengthy. Please check the number of words required by the journal.

The statements “The crystal structure analysis was performed and revealed that the molecular structure does not differ significantly. However, the crystal structures are 32 different and have been forced by C-H..O, C-H...N, and C-H…π interactions forming chain and ring 33 motifs.” Can be deleted.

Line 34 and 34. What are the binding energies of the two compounds against the PARP1 receptor? Please do state them.

Line 21 “the antiproliferative potential of synthesized compounds towards 8 anticancer cell lines.” Please do double check to ascertain the nature of the cell lines. I feel the cell line used in this study are cancer cells.

Keywords: spiropyrazoline can be used to replace crystal structure (line 40)

Introduction section

Line 44: “great attention has been focused to PARP1 inhibitors.” Replace ‘to’ with ‘on’.

Line 46: please do state some of the processes in which PARP1 is implicated with justifications.

Line 101: 8 cell lines should be replaced with 8 cancer cell lines.

The difference between compound 1 and 2 should be briefly stated.

Line 529: Cells (1×106) should be changed Cells to (1×106).

Line 540: mean SD should be replaced with mean ± SD.

Line 548: this section should come first in the method section.

Please review the journal instruction to format the ref section. MDPI has the year of publication in BOLD.

Author Response

Response to Referee

We thank the referee for positive comments and thoughtful review of the manuscript. We have made revisions according to your comments and suggestions, as described below. Changes made to the manuscript are marked in blue.

Reviewer #2:

Comment 1:

Major concerns

Results and discussion section

Response 1

The section Results and discussion was corrected and refilled according to the Reviewer's suggestions.

Comment 2: The docking results of the two compounds should be further compared with the co-crystalized ligand of the receptor or the anti-cancer drug stated previously in line 67. Furthermore, the stated energy of the complexes are the static binding energies which is often not too accurate.

I will suggest that authors carry out the dynamic binding energy of the two complexes using MM/GBSA or PB/GBSA to correct any doubt.

Is there any difference with respect to interacting atoms of the receptor between the two compounds when interacting with PARP1?

Response 2

According to the reviewer’s suggestion, the image with overlaid compounds I, II and quercetin (used as a reference PARP1 inhibitor) was added.

We agree with the reviewer that the static binding energies are often not too accurate and should be confirmed. However, we used molecular coupling calculations only to confirm that our compounds are able to reach enzyme’s active site of the enzyme. Therefore, we stated „... the potentially higher PARP1 inhibitory activity of compound I than that of compound II can be expected, however, the biological evaluation should be performed to confirm this hypothesis”. In the next sections of the manuscript we provide experimental biological data, which proved our initial calculations. Having experimental results of biological assays we did not perform more accurate (but also more time-consuming) calculations, because in our opinion the experimental results are the most appropriate confirmation of any theoretical data.

The binding modes for compounds I and II occurred to be different; for example, the interaction between the parasubstituted methyl group of compound II and hydrophobic amino acid residues was not detected – the appropriate sentence was added to the manuscript.

Comment 3: In the cytotoxicity activity section (Line 205): It is worth noting the selectivity of the compounds between normal and cancer cells. The cytotoxicity of these compounds should be evaluated against the corresponding normal cells of the studied cancer cells.

Response 3

Thank you for your valuable attention to cytotoxicity tests on normal cells. This is an extremely important aspect of biological research, especially in terms of the potential use of the studied derivatives as chemotherapists. In this article, we used eight cancer cell lines, and the experiments we conducted are of the nature of basic research in the field of basic evaluation of molecular pathways of anticancer activity. By studying the effect of compounds on PARP1, quantifying the apoptosis process, and analyzing genotoxicity, we tried to emphasize the antiproliferative nature of the spiropyrazoline derivatives. Of course, further research that we are planning provides for a significant expansion of research that will allow us to examine the anticancer properties more closely. Additionally, we plan to assess the cytotoxic effect of the analyzed compounds on normal endothelial cells. These cells will be isolated by us from umbilical cords (biological material obtained in cooperation with the Medical University). Unfortunately, due to the limited availability of biological material, such as umbilical cords, it was impossible to include these studies in this publication. However, we emphasize that due to the very promising results of this work, research on the anticancer properties of spiropyrazoline derivatives will continue and certainly extended to the mentioned normal cells.

Comment 4: For the comet assay, the result should be compared to a control.

Response 4

As shown in Figure 8, all the results from the comet DNA damage test for 8 cancer cell lines are for cells incubated for 24 h with analyzed derivatives and control cells that were not treated with the spiropyrazoline derivatives. For cancer control cells, the level of DNA damage was about 1.5-2%. Assessment of the percentage of DNA in the comet tail for control and exposed derivative cells was performed independently by Lucia-Comet v. 4.51 program, based on measurements of the color intensity and the length of the comet's tail. This program analyzes the intensity of fluorescence and on this basis calculates a series of parameters characterizing the amount of DNA damage for each cell. The percentage of DNA in the comet's "head" corresponds to the genetic material located in the nucleus, and the fluorescent region of the "tail" of the comet corresponds to the genetic material that has left the cell nucleus as a result of migration in an electric field. The most frequently used parameter for assessing DNA damage is the tail moment, i.e. the product of the percentage of DNA in the comet's "tail" and its length.

Minor comments

Abstract section

Comment 5: The abstract feels too lengthy. Please check the number of words required by the journal.

Response 5

We shortened the abstract according to “Instructions for authors”.

Comment 6: The statements “The crystal structure analysis was performed and revealed that the molecular structure does not differ significantly. However, the crystal structures are different and have been forced by C-H..O, C-H...N, and C-H…π interactions forming chain and ring  motifs.” Can be deleted.

Response 6

As it is suggested by the referee, we deleted the above statement from the abstract.

Comment 7: Line 34 and 34. What are the binding energies of the two compounds against the PARP1 receptor? Please do state them.

Response 7

Line 34 – the free energy of binding values for compounds I and II were indicated.

Comment 8: Line 21 “the antiproliferative potential of synthesized compounds towards 8 anticancer cell lines.” Please do double check to ascertain the nature of the cell lines. I feel the cell line used in this study are cancer cells.

Response 8

Thank you for this attention. Of course, there is a factual error in the article. We used 8 cancer cell lines, not as written "anticancer". The error has been corrected in the text.

Comment 9: Keywords: spiropyrazoline can be used to replace crystal structure (line 40)

Response 9

Thank you for this suggestion. We changed the keyword: crystal structure to spiropyrazoline

Introduction section

Line 44: “great attention has been focused to PARP1 inhibitors.” Replace ‘to’ with ‘on’.

Line 46: please do state some of the processes in which PARP1 is implicated with justifications.

Line 101: 8 cell lines should be replaced with 8 cancer cell lines.

The difference between compound 1 and 2 should be briefly stated.

Line 529: Cells (1×106) should be changed Cells to (1×106).

Line 540: mean SD should be replaced with mean ± SD.

Line 548: this section should come first in the method section.

Please review the journal instruction to format the ref section. MDPI has the year of publication in BOLD.

Response 9

We would like to thank the reviewer for all comments and errors. The indicated errors have been corrected. The literature has been revised as required by the journal.

Sincerely yours,

Prof. Elzbieta Budzisz

Round 2

Reviewer 1 Report

Comments have been addressed and corrected for the most part

Author Response

Dear Reviewer 1

The authors are very grateful for reviewing the manuscript. We are pleased to see our article published in the International Journal of Molecular Science. We want to thank the Editor and the Reviewers for their comments. 

Reviewer 2 Report

Authors have made some significant changes to the previously raised concerns and the final decision lies on the editor.

Author Response

The authors are very grateful for reviewing the manuscript. We are pleased to see our article published in the International Journal of Molecular Science. We want to thank the Editor and the Reviewers for their comments.